# Composite Meal-Related Behaviors and Health Indicators: Insight from Large-Scale Nationwide Data on Korean Adults

**DOI:** 10.3390/nu17182982

**Published:** 2025-09-17

**Authors:** Seongryu Bae, Hyuntae Park

**Affiliations:** 1Department of Health Sciences, The Graduate School of Dong-A University, Busan 49315, Republic of Korea; srbae@dau.ac.kr; 2Digital Healthcare Institute, Dong-A University, Busan 49315, Republic of Korea

**Keywords:** composite meal-related behaviors, breakfast consumption, eating together, eating out, health aging

## Abstract

**Objectives:** Multidimensional dietary patterns provide a more comprehensive assessment of meal-related behavior than single behaviors, yet evidence on their variation across adulthood and association with health factors relevant to healthy aging is limited. This study examined meal-related behavior pattern distribution by age and identified predictors of unhealthy patterns. **Methods:** Data from 47,332 adults aged ≥ 18 years in the Korea National Health and Nutrition Examination Survey (2014–2022) were analyzed. Participants were divided into three age groups: young adults (18–39 years), middle-aged adults (40–64 years), and older adults (≥65 years). Within each age group, participants were further classified into three categories based on their adherence to three key meal-related behaviors: High adherence (all three behaviors: regular breakfast, shared mealtimes, and infrequent eating out), Moderate adherence (two behaviors), or Low adherence (one or none). Group differences in sociodemographic, clinical variables, nutrient intake, biochemical markers, and lifestyle factors were tested using ANOVA and Pearson’s chi-square, and predictors were identified with multinomial logistic regression. **Results**: Older adults most frequently showed the High adherence group (51.1%) but had the lowest prevalence of shared mealtimes (66.3%), suggesting social vulnerabilities despite healthy individual habits. Young adults had the highest Low pattern prevalence, which was primarily driven by infrequent breakfast and frequent eating out. Across all ages, poorer patterns were consistently associated with living alone, prolonged sedentary time, and adverse metabolic profiles. Middle-aged adults exhibited moderate adherence to healthy behaviors but showed the highest prevalence of chronic disease. In older adults, poorer patterns were associated with lower intakes of energy, carbohydrate, protein, and dietary fiber intake, alongside higher rates of living alone and sedentary behavior. **Conclusions:** Composite meal-related behaviors differed across age groups as follows: Young adults most frequently exhibited low adherence, middle-aged adults showed moderate adherence but bore the highest burden of chronic diseases, while older adults demonstrated high breakfast adherence and low frequency of eating out, but faced nutritional insufficiency and social vulnerability. These findings suggest the need for personalized interventions for each age group.

## 1. Introduction

Meal-related behaviors such as regular breakfast consumption, shared mealtimes, and frequency of eating out are well-established determinants of nutritional quality and long-term health outcomes [1]. Epidemiological evidence indicates that skipping breakfast is associated with elevated cardiometabolic risk, including obesity, hypertension, unfavorable lipid profiles, diabetes, and higher mortality [2,3]. In older populations, skipping breakfast has also been linked to accelerated cognitive decline, emphasizing the broader health implications of this behavior [4]. Likewise, eating together has been positively associated with better nutritional quality and health outcomes. Evidence from adult populations indicates that frequent family meals are positively associated with higher fruit and vegetable consumption among parents. Fathers who regularly participate in shared meals report lower fast-food intake, while mothers demonstrate reduced engagement in dieting behaviors. These findings suggest that the benefits of communal eating extend to adult dietary choices and weight-related attitudes [5]. In older Japanese adults living alone, participation in shared meals has been linked to better self-rated health, healthier dietary practices, and lower levels of frailty [6]. Although eating out occasionally is a part of modern lifestyles, evidence from U.S. adult cohort and cross-sectional studies indicates that higher frequency of out-of-home meal consumption is associated with poorer overall diet quality and unfavorable patterns in chronic disease biomarkers [7]. Furthermore, findings from a large prospective cohort of 35,084 adults showed that consuming two or more meals away from home per day was linked to an increased risk of all-cause mortality [8].

Understanding meal-related behaviors across the life course is valuable for describing age-specific patterns and identifying potential areas for intervention. Although this cross-sectional study cannot establish causal relationships with healthy aging, the observed meal-related behaviors and nutritional patterns highlight important correlates of health in aging populations. Previous research has shown that adequate nutrition, social engagement during meals, and physical activity are associated with better health outcomes in older adults [9,10,11]. By examining dietary behaviors across age groups, we can identify potential areas for intervention to support healthy aging. However, previous research has predominantly examined these behaviors in isolation, limiting understanding of their combined influence on health. Studies rarely integrate multiple meal-related behaviors into a single composite indicator, and few have investigated how such patterns relate to sociodemographic, clinical, biochemical, sedentary, and nutritional characteristics in different stages of adulthood. Meal-related behaviors differ substantially across the life course, reflecting changing social roles, lifestyle constraints, and health demands. Younger adults often face competing demands from education, work, and social activities, which increase the likelihood of skipping meals, dining out, or eating alone [12,13]. Middle-aged adults may experience dietary changes due to family responsibilities, income stability, or emerging health concerns, and are more likely to consume meals away from home [14]. Older adults, conversely, tend to adhere to traditional eating patterns, such as regular breakfast consumption and reduced eating-out frequency [15,16], but they are also increasingly vulnerable to reduced social engagement during meals, particularly among those living alone. In Korea, national survey data show that breakfast skipping is highly prevalent among young adults: for example, one study reported that among 20–39-year-olds, breakfast skipping rates increased from 43.2% to 55.4% in males and from 36.6% to 63.3% in females between 2013 and 2022 [17]. Similarly, the frequency of eating out has significantly increased in recent years, especially among younger and middle-aged adults. Analysis of 2010–2015 KNHANES data found that individuals with higher eating-out frequency consumed greater amounts of energy, carbohydrates, proteins, fats, and sodium than those with lower eating-out frequency [18]. These variations suggest that the associations between meal-related behaviors and health indicators may differ substantially by age group, and that pooled analyses may obscure meaningful age-specific trends.

The Korea National Health and Nutrition Examination Survey (KNHANES) provide an ideal platform for such an investigation. Its large, nationally representative sample, comprehensive dietary and health assessments, and standardized methodology enable robust stratified analyses. Using KNHANES data allows for simultaneous evaluation of sociodemographic, clinical, biochemical, sedentary behavior, and nutrient intake variables in relation to composite meal-related behavior patterns, while maintaining sufficient statistical power across age strata.

Therefore, the present study aimed to (1) categorize adults into adherence group based on three key meal-related behaviors; (2) examine how these adherence groups are distributed across young, middle-aged, and older adults; and (3) assess the associations between meal-related behavior adherence and health-related variables within each age group.

## 2. Materials and Methods

### 2.1. Study Design and Data Source

This study was designed as a cross-sectional observational study using data from the 6th to 9th (2014–2022) rounds of the Korea National Health and Nutrition Examination Survey (KNHANES). KNHANES is a nationwide representative survey conducted exclusively in Republic of Korea by the Korea Disease Control and Prevention Agency (KDCA) under the Ministry of Health and Welfare [19]. It has contributed significantly to the development of national public health policies and programs and provides a robust dataset for academic research and international comparative studies. The survey data are publicly accessible through the official KNHANES website (https://knhanes.kdca.go.kr/knhanes/main.do, accessed on 15 June 2025). A total of 55,912 adults aged ≥ 18 years were eligible for analysis (young adults: n = 15,455; middle-aged adults: n = 25,416; older adults: n = 15,041). After excluding participants with missing information on meal-related behaviors, covariates, or outcome variables, 47,332 individuals remained in the final analytic sample (young adults: n = 12,668; middle-aged adults: n = 21,268; older adults: n = 13,396), which was used for all subsequent analyses [20,21].

The data utilized in this study were approved by the Institutional Review Board of the Korea Centers for Disease Control and Prevention under the following approval numbers: 2013-07CON-03-4C, 2013-12EXP-03-5C, 2018-01-03-P-A, 2018-01-03-C-A, 2018-01-03-2C-A, 2018-01-03-5C-A, and 2018-01-03-4C-A. All participants provided written informed consent prior to participation.

### 2.2. Categorization Based on Dietary Behaviors

Participants were classified into three groups reflecting their adherence to meal-related behaviors (breakfast consumption [2,3,4], eating together [5,6], and frequency of eating out [7,8]), which have been shown in previous epidemiological studies to be important predictors of nutritional quality and long-term health outcomes. Breakfast consumption was assessed by asking participants about their average weekly breakfast frequency over the past year. Participants who consumed breakfast three or more days per week were classified as “regular consumers,” while those who consumed breakfast two or fewer times per week were classified as “breakfast skippers.” Eating together was assessed by asking participants whether they had eaten dinner with family or others over the past year. The frequency of eating out was defined as the average number of times participants reported eating outside the home over the past year, with “frequent” defined as two or more times per week and “rare” defined as less than once per month.

Individuals who engaged in all three healthy behaviors were categorized into the High adherence group. Those meeting two of the three criteria were classified as the Moderate adherence group, and those meeting one or none were placed in the Low adherence group.

### 2.3. Sociodemographic Characteristics and Clinical Variables

Sociodemographic characteristics included age (years), sex (female), living arrangement (living alone: yes/no), and Basic Livelihood Security recipient status (yes/no), as reported in the health interview survey. Hypertension (yes/no), dyslipidemia (yes/no), diabetes mellitus (yes/no) were assessed using self-reported questionnaires. Body mass index (BMI) was calculated as body weight (kg) divided by the square of height (m). We categorized participants according to the Asian—Pacific guidelines [22], which define BMI groups as underweight (<18.5 kg/m^2^), normal (18.5–22.9 kg/m^2^), overweight (23.0–24.9 kg/m^2^), and obese (≥25.0 kg/m^2^).

### 2.4. Assessment of Sedentary Time

Sedentary behavior was assessed using a self-reported questionnaire. Participants were asked the following question: “How much time do you spend sitting each day?” This includes time spent at work, at home, while doing course work and during leisure time. This may include time spent sitting at a desk, visiting friends, reading, or sitting or lying down to watch television. The average daily sedentary time (in hours per day) was recorded based on participants’ estimates.

### 2.5. Biochemical Markers

Venous blood samples were collected from the participants and immediately processed, refrigerated, and transported to the central laboratory of the Seegene Medical Foundation in Seoul. Biochemical analyses, including measurements of fasting plasma glucose, total cholesterol, and triglycerides, were performed using the enzymatic methods on an automated analyzer (Hitachi Automatic Analyzer 7600, Tokyo, Japan).

### 2.6. Assessment of Nutrient Intake

Nutrient intake information was obtained through in-person interviews conducted by trained personnel, including dietitians. A single 24-h dietary recall was used to capture all foods and beverages consumed by participants during the previous day, along with details on portion sizes and preparation methods. Nutrient and energy intakes—including total calories, carbohydrates, protein, fat, cholesterol, saturated fatty acid, and dietary fiber—were estimated using the Can-Pro 2.0 software (Korean Dietetic Association, Seoul, Republic of Korea) [23], a validated tool commonly utilized in clinical and epidemiological nutrition research in Korea. This software contains an extensive database of Korean foods and their nutrient compositions. The resulting data served as the basis for evaluating participants’ dietary patterns and nutrient intake profiles.

### 2.7. Statistical Analysis

Continuous variables were presented as means and standard deviations (SD), and categorical variables were expressed as counts and percentages. The normality of continuous variables was assessed using the Kolmogorov–Smirnov test. Given the large sample sizes in each group, minor deviations from normality were expected. Visual inspection of histograms and Q–Q plots confirmed approximate normality. In addition, according to the central limit theorem (CLT), the sampling distribution of the mean approximates normality as the sample size increases, even when the underlying population distribution is not strictly normal [24,25]. Therefore, parametric tests were deemed appropriate. To compare variables across the three meal-related behavior groups, one-way analysis of variance (ANOVA) with post hoc Bonferroni correction was conducted for continuous variables, and chi-square (χ^2^) tests were used for categorical variables. Adjusted standardized residuals were calculated to determine specific group differences within categorical variables in the χ^2^ test. A cell was considered to have significantly more participants than expected when the adjusted residual exceeded 1.96, and significantly fewer if it was below −1.96. To evaluate age group differences in the distribution of dietary behavior patterns, 100% stacked bar charts were constructed. In addition, a radial bar chart was employed to visualize the proportion of each age group engaging in healthy meal-related behaviors, including regular breakfast consumption, frequent eating together, and infrequent eating out. To examine the associations between dietary behavior status (reference: High adherence group) and sociodemographic, clinical, biochemical marker, sedentary behavior, and nutrient intake status, multinomial logistic regression analyses were conducted. Odds ratios (ORs) and 95% confidence intervals (CIs) were estimated. Analyses were conducted using SPSS version 28.0 (IBM Corp., Armonk, NY, USA). Statistical significance was set a priori at *p* < 0.05.

## 3. Results

### 3.1. Differences in Dietary Behaviors Across Age Groups

Figure 1a shows the age-specific differences between groups categorized as High, Moderate, and Low according to their adherence to three meal-related behaviors (not skipping breakfast, eating together, and rarely eating out). The High adherence group, indicating adherence to all three behaviors, was most prevalent among older adults, followed by middle-aged adults, and least common in young adults. Conversely, the Low adherence group reflecting adherence to one or none of the healthy behaviors, was most frequent in young adults, decreased in middle-aged adults, and was rare among older adults. These findings suggest a trend in which healthy meal-related behaviors become increasingly prevalent with age. While older adults tended to engage more consistently in beneficial eating practices, younger adults appear more likely to skip meals, eat out frequently, or dine alone—behaviors that may reflect lifestyle constraints or lower health awareness.

Figure 1b presents the dietary characteristics of young adults, middle-aged adults, and the older adults. Older adults exhibited the highest proportion of not skipping breakfast (95.3%), followed by middle-aged adults (77.2%) and young adults (50.9%). This suggests that maintaining a regular breakfast habit becomes more prevalent with age. Similarly, the behavior of rarely eating out was most common among older adults (90.0%), indicating a preference for home-prepared meals or more traditional eating patterns in this age group. Middle-aged adults followed at 58.0%, and young adults reported the lowest rate (44.7%). In contrast, the pattern for eating together showed an inverse age trend. Young adults reported the highest prevalence (81.2%), followed by middle-aged adults (79.2%), while older adults were least likely to report shared meals (66.3%). This may reflect social or environmental factors, such as household composition, living alone, or reduced social engagement among older individuals. Overall, while older adults demonstrated more consistent adherence to individual-based healthy eating behavior such as breakfast consumption and avoiding eating out, younger adults appeared to be more socially engaged in shared meal settings.

### 3.2. Comparison of Health Outcomes by Meal-Related Behavior Groups

Table 1, Table 2 and Table 3 present the comparison of sociodemographic, clinical, biochemical markers, sedentary behavior and nutritional characteristics according to meal-related behavior groups among young adults, middle-aged adults, and older adults, respectively. Across young, middle-aged, and older adults, several consistent patterns emerged with respect to meal-related behavior groups. In all three age groups, BMI increased progressively from the High adherence to the Low adherence group (all *p* < 0.001). Similarly, BMI category distributions showed that participants in the High adherence group were more likely to have normal weight, whereas the prevalence of obesity increased toward the Low adherence group across all age groups. For example, in young adults, obesity rose from 27.4% in the High adherence group to 33.2% in the Low adherence group, and in middle-aged adults from 35.5% to 39.8%. A comparable pattern was also observed among older adults, indicating a consistent gradient across age groups. The prevalence of living alone (all *p* < 0.001) were highest in the Low adherence group. Among middle-aged and older adults, the proportion of recipients of the Basic Livelihood Security Program increased progressively from the High adherence to the Low adherence group (all *p* < 0.001), with this socioeconomic gradient being more prominent in the older adults. Unlike other age groups, middle-aged adults exhibited a progressive increase in the prevalence of clinical conditions such as hypertension (*p* < 0.001), dyslipidemia (*p* < 0.001), and diabetes mellitus (*p* = 0.027) across dietary groups, from the High adherence to the Low adherence group. Sedentary time was longest in the Low adherence group and shortest in the High adherence group, regardless of age (all *p* < 0.001). Metabolic markers, including fasting glucose, total cholesterol, and triglycerides, were consistently worse in the Low adherence group than in the High adherence group in all age groups.

Across all age groups, biochemical markers, including fasting glucose and total cholesterol were consistently worse in the Low adherence group than in the High adherence group. A similar pattern was observed for triglycerides in all age groups except older adults. Nutrient intake also differed significantly across meal-related behavior groups. In the young adults, except for carbohydrate intake, the High adherence group exhibited the lowest consumption of total energy (*p* < 0.001), fat (*p* < 0.001), cholesterol (*p* < 0.001), and saturated fat (*p* < 0.001), while the highest intake levels were consistently observed in the Low adherence group. However, Protein and dietary fiber intake were highest in the High adherence group and lowest in the Low adherence group (all *p* < 0.001). A similar pattern was observed among middle-aged adults, with higher intake of total energy, fat, cholesterol, and saturated fatty acids, and lower intake of dietary fiber in the Low adherence group compared to the High adherence group (all *p* < 0.001). Unlike younger age groups, older adults exhibited a distinct pattern in which total energy (*p* < 0.001), carbohydrate (*p* < 0.001), and protein (*p* < 0.001) intake decreased toward the Low adherence group. In contrast, intake of cholesterol (*p* = 0.017) and saturated fatty acids (*p* = 0.003) was higher in the Low adherence group than in the Moderated adherence group. Consistent with other age groups, dietary fiber intake was lowest in the Low adherence group (*p* < 0.001). These findings reveal important nutritional challenges in older adults. The combination of these factors in the Low adherence group—particularly low protein intake (53.1 ± 30.5 g), prolonged sedentary time (9.0 ± 4.0 h/day), and social isolation (52.7% living alone)—suggests potential vulnerabilities that warrant attention in nutrition interventions for older adults.

### 3.3. Predictors of Poorer Dietary Status

Results of the multinomial logistic regression analyses, stratified by age group, are presented in Figure 2 and Appendix A.

In young adults (Figure 2a and Appendix A), males were more likely to belong to the Moderate adherence (OR = 2.02, 95% CI: 1.69–2.41) and Low adherence groups (OR = 2.59, 95% CI: 2.14–3.12) compared to females. Living alone was associated with higher odds of being in the Moderate adherence (OR = 1.82, 95% CI: 1.39–2.39) and Low adherence groups (OR = 2.46, 95% CI: 2.01–3.57). Sedentary time was significantly longer in the Low adherence groups (OR = 1.20, 95% CI: 1.08–1.11), and fasting glucose was higher (OR = 1.71, 95% CI: 1.53–1.91). In terms of nutrient intake, the Low adherence groups exhibited higher total energy intake (OR = 1.43, 95% CI: 1.28–1.61), protein intake (OR = 1.99, 95% CI: 1.69–2.13), fat intake (OR = 1.55, 95% CI: 1.39–1.93), and lower dietary fiber intake (OR = 0.65, 95% CI: 0.55–0.71) than the High adherence groups.

Among middle-aged adults (Figure 2b and Appendix A), males were also more likely to be in the Good (OR = 2.29, 95% CI: 2.07–2.53) and Low adherence groups (OR = 3.08, 95% CI: 2.71–3.50). Living alone was strongly associated with both Moderate adherence (OR = 2.89, 95% CI: 1.99–3.88) and Low adherence groups (OR = 3.12, 95% CI: 2.57–4.01). Clinical characteristics showed that hypertension (OR = 1.81, 95% CI: 1.65–2.59), dyslipidemia (OR = 2.13, 95% CI: 1.78–2.98), and diabetes mellitus (OR = 1.62, 95% CI: 1.41–1.88) were significantly more prevalent in the Low adherence groups. In the sedentary time, compared to the High adherence groups, middle-aged adults in the Low adherence groups had 1.57 times higher odds of prolonged sedentary time (95% CI: 1.43–1.78). In the biochemical marker, the Low adherence groups had higher total cholesterol (OR = 2.11, 95% CI: 1.78–3.11), triglycerides (OR = 2.25, 95% CI: 1.82–3.41). Nutrient intake analysis showed that the Low adherence groups had higher total energy intake (OR = 1.12, 95% CI: 1.09–1.29), fat intake (OR = 2.31, 95% CI: 1.95–2.91), cholesterol intake (OR = 2.29, 95% CI: 1.82–3.08), and saturated fatty acid intake (OR = 1.86, 95% CI: 1.74–2.39), as well as lower dietary fiber intake (OR = 0.77, 95% CI: 0.69–0.81).

In older adults (Figure 2c and Appendix A), males were also more likely to be in the Low adherence groups (OR = 1.56, 95% CI: 1.23–1.98). living alone was a particularly strong predictor of poorer dietary status, with markedly higher odds for both Moderate (OR = 3.11, 95% CI: 2.81–3.74) and Low adherence groups (OR = 4.35, 95% CI: 4.09–5.02). Older adults in the Basic Livelihood Security Program were strongly associated with poorer dietary behavior status, compared to the High adherence groups (OR = 2.11, 95% CI: 1.63–2.74). However, no statistically significant associations were observed between meal-related behavior status and clinical characteristics such as hypertension, dyslipidemia, and diabetes mellitus. Sedentary time was longer in the Low adherence groups (OR = 1.54, 95% CI: 1.41–1.66). The Low adherence groups also had higher total cholesterol (OR = 1.71, 95% CI: 1.64–1.92) and lower energy (OR = 0.79, 95% CI: 0.61–0.86), carbohydrate (OR = 0.84, 95% CI: 0.79–0.91), protein intake (OR = 0.86, 95% CI: 0.79–0.96), and dietary fiber intake (OR = 0.71, 95% CI: 0.67–0.89).

## 4. Discussion

This nationally representative study provides comprehensive insights into meal-related behaviors across adulthood in Korea. Our findings reveal an interesting paradox: while older adults demonstrated the highest adherence to traditionally ”good” meal-related behaviors such as regular breakfast consumption and infrequent eating out, they also faced unique nutritional and social challenges that may impact their health and well-being. This complexity highlights the need for nuanced, age-specific approaches to dietary interventions [15,26]. A more detailed analysis showed that within these composite patterns, the prevalence of individual behaviors varied greatly depending on age. Older adults were much more likely to eat breakfast regularly and rarely eat out, but were less likely to eat meals together with others than younger and middle-aged adults. The high adherence of older adults to regular breakfast consumption (95.3%) and infrequent eating out (90.0%) is consistent with prior studies showing that older populations tend to maintain traditional, home-based eating routines, due to established habits, more time for meal preparation, and heightened health awareness [15,16]. In contrast, young adults often face competing demands from education, employment, and social activities, which have been linked to higher rates of breakfast skipping and reliance on convenience foods or meals away from home [12,13].

The prevalence of shared mealtimes was lowest among older adults (66.3%), consistent with research findings that aging is often accompanied by reduced opportunities for social eating due to widowhood, household size reduction, and social isolation [27,28]. Reduced social engagement during meals has been associated with poorer dietary variety, increased risk of malnutrition, and greater frailty [6,27]. Therefore, while older adults may have favorable dietary habits such as regular breakfast intake or less frequent eating out, low frequency of shared meals may indicate significant social and nutritional vulnerabilities in this group. Beyond these social factors, economic constraints and limited access to community-based or age-friendly dining environments may further restrict opportunities for shared meals. To address these challenges, targeted interventions—such as community meal programs, intergenerational dining initiatives, and integration of nutritional support with social engagement strategies—could be effective in mitigating both nutritional inadequacy and social isolation among older adults. The inverse pattern observed in young adults—low frequency of breakfast and home-prepared meals but high frequency of shared meals (81.2%)—is estimated to be due to social relationships rather than health-conscious behaviors such as sharing meals with colleagues or family members or eating with colleagues or friends. In this population, eating together may also be associated with high frequencies of eating out. Therefore, in environments where frequent consumption of energy-dense foods or ultra-processed foods occurs, this pattern may not be consistent with improvements in meal quality [29]. Middle-aged adults were in the middle position in all three behaviors, suggesting that this stage of life may be an important time for intervention. Family caregiving and increasingly prevalent chronic diseases may be factors that keep adherence to healthy eating habits at an intermediate level, but work-related time constraints may still limit optimal eating patterns [14]. This study, adopting a life stages approach, provides evidence that public health policies should consider both the nutritional and social aspects of dietary habits to improve health outcomes throughout the adult life span. The age-related patterns observed in this study provide important context for understanding meal-related behaviors as potential modifiable factors for healthy aging. Young adults’ poor meal-related behaviors, while concerning for immediate health, also represent missed opportunities for establishing healthy habits that could benefit long-term health. The intermediate patterns in middle-aged adults, coinciding with higher chronic disease prevalence, suggest this may be a critical period for intervention.

Comparison of groups with high, moderate, and low adherence behaviors showed that, across all age groups, groups with poor nutritional status had higher BMI, were more likely to live alone, and had consistently negative metabolic profiles, including higher fasting blood glucose, total cholesterol, and triglyceride levels. This finding is consistent with previous studies indicating that poor dietary behavior is associated with increased cardiovascular metabolic risk and chronic disease [23,28,30]. These results extend our findings by showing that adherence to meal-related behaviors is not only associated with lower mean BMI but also with healthier BMI distributions. Participants with higher adherence to breakfast consumption, shared meals, and infrequent eating out were more likely to maintain normal weight, whereas those with poorer adherence showed greater prevalence of obesity. This pattern was consistent across young, middle-aged, and older adults, reinforcing the importance of meal-related behaviors as potential intervention targets for obesity prevention and weight management in the Korean population. The results of the nutrient intake pattern analysis showed that young adults and middle-aged adults with poor nutritional status had higher energy, fat, cholesterol, and saturated fat intake and lower dietary fiber intake, which was consistent with the high energy density and poor nutrient intake reported in previous population-based studies [31]. In contrast, older adults with poor nutritional status showed lower total energy, carbohydrate, and protein intake, suggesting potential risks of malnutrition and sarcopenia [26]. This is an age-specific pattern that has been less reported in previous studies and represents a new contribution of this study. These findings emphasize the need to address not only excessive energy intake and poor nutrient quality in younger populations but also energy and protein intake deficiencies in vulnerable older adults, while also considering the broader social and economic determinants of meal-related habits. In addition, compared with nutrient- or food-based indices such as the Healthy Eating Index [16], the Alternate Healthy Eating Index and the DASH score [32], or the Mediterranean Diet Score [33], which primarily emphasize dietary quality from a compositional perspective, our composite measure integrates meal-related behaviors (e.g., breakfast consumption, eating together, and eating out frequency) that also capture the social and behavioral dimensions of diet. This complementary perspective highlights how our approach can identify life stage–specific vulnerabilities. In particular, it emphasizes the social isolation and protein inadequacy observed in older adults, which may not be fully addressed by traditional diet quality indices. These results highlight that interventions must address not only excess energy and poor nutrient quality in younger populations but also inadequate energy and protein intake in vulnerable older adults, alongside the broader social and economic determinants of diet.

To identify factors associated with poor meal-related behavior, we used multinomial logistic regression analysis to set the High adherence group as the reference group. This analysis identified common predictors associated with poorer meal-related behavior and age-specific predictors, revealing important behavioral and nutritional deficits across the life course. In all age groups, male sex and living alone were consistent demographic predictors of poorer nutritional status. This is consistent with previous studies showing that men are less likely than women to adhere to healthy eating habits, which is thought to be due to lower nutritional knowledge, lack of cooking skills, and high dependence on convenience foods [34,35]. The association between living alone and poorer meal-related behavior has been widely reported, with particular emphasis on the role of social isolation and reduced motivation to prepare meals in contributing to poorer meal quality among older adults [28,36]. Our findings extend previous studies by showing that living alone is a strong predictor of poorer nutritional status among young and middle-aged adults. This suggests that living alone may be a priority target for nutritional interventions throughout adulthood.

Time spent sitting was another strong predictor of Lower nutritional behavior across all age groups. A sedentary lifestyle has been associated with unhealthy eating habits such as excessive consumption of energy-dense snacks and irregular meal times, which may be mediated by behavioral patterns of eating disorders, physical inactivity, and reduced meal regularity [37,38]. This dual risk pattern—low levels of physical activity combined with Low dietary habits—has been shown to synergistically increase cardiometabolic risk [38], emphasizing the need for integrated behavioral interventions. Our findings are consistent with previous research demonstrating that prolonged sedentary time is related to irregular meal timing, higher consumption of snacks, and lower overall diet quality [39,40]. Moreover, large-scale cohort studies have shown that excessive sitting contributes to obesity, type 2 diabetes, cardiovascular disease, and premature mortality [38,41]. Importantly, these risks appear to be amplified when sedentary behavior co-occurs with Low dietary patterns, suggesting a synergistic effect on adverse metabolic outcomes [42]. Together, this evidence and our findings highlight the importance of developing interventions that simultaneously target reductions in sedentary time and improvements in dietary behaviors across the lifespan.

Age-specific patterns were also observed. In middle-aged adults with poor nutritional status, the prevalence of hypertension, dyslipidemia, and diabetes was high, and total cholesterol and triglyceride levels were elevated. These results are consistent with previous studies indicating that middle age is a critical period when cumulative exposure to lifestyle factors leads to clinical diseases [32,43]. The poor dietary habits of this group were characterized by high intake of total energy, fat, cholesterol, and saturated fatty acids, and low intake of dietary fiber, a combination known to increase cardiovascular metabolic risk [33]. These findings emphasize the potential benefits of individualized dietary interventions to delay or prevent the progression of chronic diseases in middle age. On the other hand, older adults with poor dietary habits had lower total energy, carbohydrate, and protein intake, reduced dietary fiber intake, and higher total cholesterol levels. This suggests other nutritional deficiencies characterized by overall insufficient nutrient intake rather than excessive intake of unhealthy nutrients. Previous studies have shown that nutritional and protein deficiencies in the older adult population contribute to sarcopenia and frailty [44,45]. The identification of reduced protein intake as a strong independent predictor of poor dietary status in older adults is a novel contribution of this study, highlighting the need for targeted protein-focused nutritional strategies in aging populations. Furthermore, The high proportion of Basic Livelihood Support recipients among the elderly with poor dietary habits points to economic constraints as a major cause of nutrient intake deficiencies in this population [46].

The results of this study provide important implications for public health policy and clinical nutrition practice. By identifying age-specific meal-related behaviors and their sociodemographic and health-related factors, this study emphasizes the need for personalized dietary interventions that consider both nutritional quality and the social aspects of eating. For older adults, the focus should be on maintaining beneficial behaviors such as regular breakfast intake and limiting eating out, while addressing social isolation, which contributes to low frequency of eating together. For young adults, strategies are needed to reduce breakfast skipping and excessive eating out, while promoting healthy eating behaviors by leveraging existing shared meal practices. For middle-aged adults with a high burden of chronic diseases and intermediate patterns, workplace and home-based integrated interventions that overcome time constraints and promote balanced eating behaviors may be most effective. Nevertheless, these implications should be interpreted with caution given the cross-sectional and self-reported nature of our data. Future longitudinal and interventional research is required to confirm causal pathways and to determine whether targeted modifications in these behaviors can translate into tangible health benefits before informing formal policy or clinical guidelines.

The main strength of this study is that it was able to conduct robust age-specific analyses reflecting dietary trends among the adult population in Korea by utilizing a large, nationally representative dataset. Additionally, by analyzing complex meal-related behaviors rather than single behaviors, the study provided a more comprehensive understanding of how multiple lifestyle factors co-occur and interact with health outcomes. For older adults in particular, the findings suggest that promoting healthy aging requires attention to both the quality and social context of eating, highlighting the potential effectiveness of comprehensive, multi-component interventions. However, several limitations are acknowledged. First, the cross-sectional study design limits the possibility of identifying causal relationships between meal-related behaviors and health outcomes. Because our analysis is cross-sectional, causal relationships between composite meal-related behaviors and health outcomes cannot be inferred. Future longitudinal cohort studies and intervention-based trials are required to elucidate whether modifications in specific dietary behaviors lead to therapeutic benefits or expose individuals to potential adverse effects, such as nutrient deficiencies or excessive intake of unhealthy food components. Such evidence would be critical to inform dietary guidelines and targeted public health interventions. Second, the use of a single 24-h dietary recall and self-reported behaviors may limit precision and generalizability. Future research should replicate these findings in other national cohorts and longitudinal studies to confirm their external validity and to determine whether similar associations exist in populations with different cultural and dietary contexts. Third, this study was limited to Korean adults, and the results may not be directly applicable to populations with different cultural or dietary contexts. Fourth, while our study identified several factors that have been associated with health outcomes in aging populations in previous research, we did not directly assess functional status, cognitive function, or other aging-related outcomes. In addition, psychosocial determinants such as depression and social support, which may serve as important mediators or moderators of dietary behaviors, were not assessed. Future research should complement population-based surveys with qualitative assessments, functional status evaluations, and biomarker-based nutritional profiling to validate and expand upon our findings, thereby providing deeper insights into the mechanisms underlying nutritional and social vulnerabilities among older adults and informing more precise intervention strategies. Fifth, other parameters that are known to influence eating behavior, such as physical activity levels, psychosocial stress, and broader socioeconomic context, were not comprehensively assessed in our analysis. The absence of these variables may have influenced the associations we observed between meal-related behaviors and health outcomes. Therefore, interpretations regarding the implications of our findings for healthy aging should be made with appropriate caution.

To promote healthy aging, it is very important to establish appropriate dietary habits from early adulthood. Therefore, future studies should investigate the continuous association between meal-related behaviors and subsequent health outcomes, including long-term follow-up studies starting from early life. In addition, intervention studies are needed to evaluate the effectiveness of age-specific strategies aimed at improving dietary habits from nutritional and social perspectives.

## 5. Conclusions

This cross-sectional analysis of meal-related behaviors among Korean adults reveals important age-specific differences and challenges. Young adults were most affected by time-constrained behaviors such as breakfast skipping and frequent eating out, whereas older adults faced nutritional inadequacy and social isolation despite regular meal consumption. In particular, older adults in the Low adherence group showed insufficient protein intake below recommended levels for maintaining muscle mass, combined with prolonged sedentary behavior and a high prevalence of living alone, indicating a complex profile of modifiable vulnerabilities associated with frailty risk. The paradoxical finding that older adults maintained regular meals but reported the lowest prevalence of shared mealtimes underscores the need to address both nutritional adequacy and the social context of eating.

These findings emphasize the importance of a multidimensional, age-specific approach that integrates strategies to improve nutritional quality and strengthen the social dimension of meals. Public health policies and community programs should be tailored to each age group, focusing on promoting healthy individual dietary habits among young adults, supporting balanced meal practices in middle-aged adults, and addressing malnutrition and social isolation in older adults. While causal relationships cannot be inferred from this cross-sectional design, the study provides a foundation for future longitudinal and interventional research to develop targeted strategies supporting healthy dietary behaviors and, ultimately, healthy aging in the Korean population.

## Figures and Tables

**Figure 1 nutrients-17-02982-f001:**
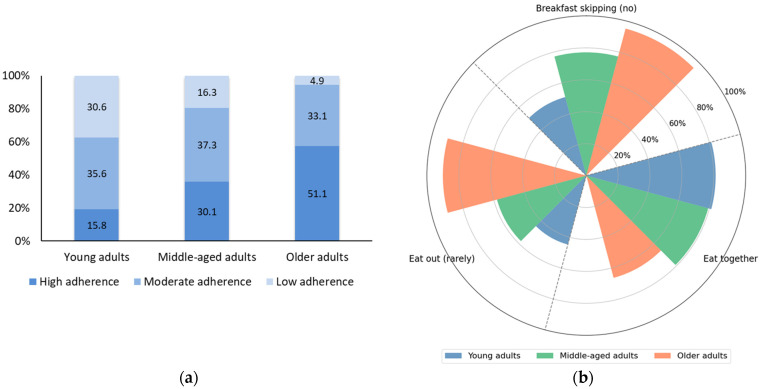
Distribution of composite meal-related behaviors and meal-related behaviors by age group. (**a**) Proportion of participants classified into High, Moderate, and Low adherence groups based on meal-related behaviors across young, middle-aged, and older adults. (**b**) Prevalence of adherence to each individual meal-related behavior—regular breakfast consumption, infrequent eating out, and eating together—across age groups.

**Figure 2 nutrients-17-02982-f002:**
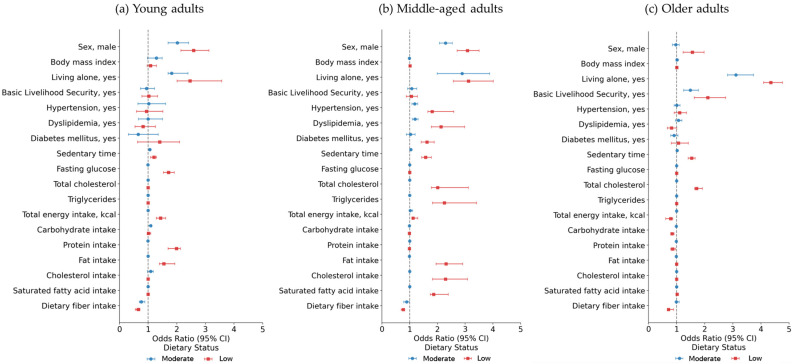
Forest plots of associations between meal-related behavior groups (Moderate and Low) and sociodemographic, clinical, biochemical, sedentary, and nutritional variables, stratified by age group. Odds ratios (OR) and 95% confidence intervals (CI) are shown for each variable, with the reference group being the High adherence group. (**a**) Young adults (18–39 years); (**b**) Middle-aged adults (40–64 years); (**c**) Older adults (≥65 years).

**Table 1 nutrients-17-02982-t001:** Comparison of characteristics among young adults by meal-related behavior group.

Variable	Very Good (n = 2436)	Good (n = 5501)	Poor (n = 4731)	*p*-Value
Age, year	31.4 ± 6.2 ^a,b^	29.5 ± 6.6 ^c^	29.0 ± 6.2	<0.001 ^#^
Sex, female	1787 (73.4) ^d^	3023 (55.0)	2188 (46.2) ^e^	<0.001 ^$^
BMI, kg/m^2^	23.1 ± 4 ^b^	23.3 ± 4.2 ^c^	23.7 ± 4.3	<0.001 ^#^
BMI group, n (%)				<0.001 ^$^
Underweight	184 (8.2)	390 (7.8)	277 (6.5) ^e^	
Normal	1067 (47.5) ^d^	2252 (45.3)	1813 (42.6)	
Overweight	381 (17.0)	829 (16.7)	754 (17.7)	
Obese	615 (27.4) ^e^	1499 (30.2)	1415 (33.2) ^d^	
Living alone, yes	40 (1.6) ^e^	237 (4.3) ^e^	805 (17.0) ^d^	<0.001 ^$^
Basic Livelihood Security, yes	115 (4.7)	256 (4.7)	254 (5.4)	0.215 ^$^
Hypertension, yes	31 (1.4)	86 (1.8)	92 (2.2)	0.049 ^$^
Dyslipidemia, yes	40 (1.8)	101 (2.1)	103 (2.5)	0.153 ^$^
Diabetes mellitus, yes	14 (0.6)	38 (0.8)	31 (0.7)	0.773 ^$^
Sedentary time, hour/day	8.2 ± 3.5 ^a,b^	9.0 ± 3.5 ^c^	9.5 ± 3.6	<0.001 ^#^
Fasting glucose, mg/dL	91.8 ± 16.7 ^b^	92.1 ± 13.8 ^c^	93.5 ± 15.9	<0.001 ^#^
Total cholesterol, mg/dL	184.7 ± 34.0 ^b^	185.6 ± 34.0 ^c^	187.7 ± 33.5	<0.001 ^#^
Triglycerides, mg/dL	105.5 ± 85.9 ^a,b^	112.8 ± 92.2 ^c^	121.1 ± 107.7	<0.001 ^#^
Total energy intake, kcal	1977.4 ± 873.0 ^a,b^	2102.4 ± 992.7 ^c^	2170.7 ± 1127.4	<0.001 ^#^
Carbohydrate intake, g	284.1 ± 121.5	286.1 ± 126.6	280.6 ± 128.3	0.09 ^#^
Protein intake, g	75.0 ± 42.1 ^a,b^	79.6 ± 46.1 ^c^	82.1 ± 52.1	<0.001 ^#^
Fat intake, g	52.5 ± 35.3 ^a,b^	58.8 ± 41.3 ^c^	62.9 ± 51.1	<0.001 ^#^
Cholesterol intake, mg	292.8 ± 246.3 ^b^	303.2 ± 246 ^c^	315.8 ± 274.9	<0.001 ^#^
Saturated fatty acid intake, g	16.8 ± 12.5 ^a,b^	19.0 ± 14.6 ^c^	20.5 ± 18.2	<0.001 ^#^
Dietary fiber intake, g	22.5 ± 12.3 ^b^	21.8 ± 12 ^c^	20.6 ± 11.6	<0.001 ^#^

Values are presented as mean ± SD or n (%). ^#^, *p*-values for continuous variables were obtained using one-way ANOVA. Superscript letters denote post hoc comparisons: ^a^, high vs. moderate; ^b^, high vs. low; ^c^, moderate vs. low. ^$^, *p*-values for categorical variables were obtained using Pearson’s chi-square test. ^d^, Statistically significant association by adjusted standardized residual > 1.96 (*p* < 0.05). ^e^, Statistically significant association by adjusted standardized residual < −1.96 (*p* < 0.05).

**Table 2 nutrients-17-02982-t002:** Comparison of characteristics among middle-aged adults by meal-related behavior group.

Variable	Very Good (n = 7648)	Good (n = 9471)	Poor (n = 4149)	*p*-Value
Age, year	54.0 ± 7.3 ^a,b^	51.9 ± 7 ^c^	50.2 ± 6.8	<0.001 ^#^
Sex, female	5498 (71.9) ^d^	4875 (51.5)	1909 (46.0) ^e^	<0.001 ^$^
BMI, kg/m^2^	24.0 ± 3.4 ^a,b^	24.2 ± 3.4	24.3 ± 3.5	<0.001 ^#^
BMI group, n (%)				<0.001 ^$^
Underweight	174 (2.4)	210 (2.4)	96 (2.5)	
Normal	2779 (38.7) ^d^	3108 (35.8)	1307 (34.2) ^e^	
Overweight	1676 (23.3)	2100 (24.2)	898 (23.5)	
Obese	2553 (35.5) ^e^	3272 (37.7)	1519 (39.8) ^d^	
Living alone, yes	167 (2.2) ^e^	862 (9.1)	785 (18.9) ^d^	<0.001 ^$^
Basic Livelihood Security, yes	394 (5.2) ^e^	568 (6.0)	301 (7.3) ^d^	<0.001 ^$^
Hypertension, yes	1644 (18.7) ^e^	1810 (20.6)	727 (22.7) ^d^	<0.001 ^$^
Dyslipidemia, yes	1724 (17.7) ^e^	1766 (20.1)	689 (23.8) ^d^	<0.001 ^$^
Diabetes mellitus, yes	655 (7.5)	760 (8.7)	293 (9.0) ^d^	0.027 ^$^
Sedentary time, hour/day	7.3 ± 3.3 ^a,b^	7.8 ± 3.5 ^c^	8.3 ± 3.6	<0.001 ^#^
Fasting glucose, mg/dL	101.9 ± 23.5 ^a,b^	102.8 ± 24.3	103.8 ± 27.3	<0.001 ^#^
Total cholesterol, mg/dL	196.3 ± 38.3 ^b^	197.1 ± 37.2 ^c^	199.8 ± 38.3	<0.001 ^#^
Triglycerides, mg/dL	132.5 ± 95.0 ^a,b^	145.6 ± 123.0 ^c^	151.3 ± 125.2	<0.001 ^#^
Total energy intake, kcal	1860.1 ± 759.7 ^a,b^	2025.9 ± 895.7	1989.7 ± 900.6	<0.001 ^#^
Carbohydrate intake, g	293.2 ± 122.2 ^a,b^	300.5 ± 124.9 ^c^	282.5 ± 119.6	<0.001 ^#^
Protein intake, g	66.8 ± 32.3 ^a,b^	73.4 ± 43.4	71.6 ± 37.9	<0.001 ^#^
Fat intake, g	40.2 ± 27.7 ^a,b^	45.8 ± 32.2 ^c^	47.5 ± 34.7	<0.001 ^#^
Cholesterol intake, mg	224.0 ± 198.1 ^a,b^	254.2 ± 235.1	256.8 ± 226.6	<0.001 ^#^
Saturated fatty acid intake, g	12.1 ± 9.5 ^a,b^	14.0 ± 10.8 ^c^	14.8 ± 11.9	<0.001 ^#^
Dietary fiber intake, g	28.4 ± 15.0 ^a,b^	27.8 ± 14.3 ^c^	25.0 ± 13.0	<0.001 ^#^

Values are presented as mean ± SD or n (%). ^#^, *p*-values for continuous variables were obtained using one-way ANOVA. Superscript letters denote post hoc comparisons: ^a^, high vs. moderate; ^b^, high vs. low; ^c^, moderate vs. low. ^$^, *p*-values for categorical variables were obtained using Pearson’s chi-square test. ^d^, Statistically significant association by adjusted standardized residual > 1.96 (*p* < 0.05). ^e^, Statistically significant association by adjusted standardized residual < −1.96 (*p* < 0.05).

**Table 3 nutrients-17-02982-t003:** Comparison of characteristics among older adults by meal-related behavior group.

Variable	Very Good (n = 7690)	Good (n = 4976)	Poor (n = 730)	*p*-Value
Age, year	72.9 ± 5.0 ^a,b^	73.5 ± 5.1 ^c^	71.9 ± 5.2	<0.001 ^#^
Sex, female	3949 (51.4)	3240 (65.1) ^d^	412 (56.4)	<0.001 ^$^
BMI, kg/m^2^	24.0 ± 3.2 ^a,b^	24.2 ± 3.2	24.4 ± 3.3	<0.001 ^#^
BMI group, n (%)				0.041
Underweight	208 (2.9)	117 (2.5)	25 (3.7)	
Normal	2499 (36.1) ^d^	1519 (34.0)	240 (31.3)	
Overweight	1802 (23.9)	1173 (25.4)	179 (25.8)	
Obese	2617 (37.1)	1801 (38.1)	235 (39.2)	
Living alone, yes	250 (3.3) ^e^	2482 (49.9)	385 (52.7) ^d^	<0.001 ^$^
Basic Livelihood Security, yes	518 (6.7) ^e^	767 (15.4)	152 (20.9) ^d^	<0.001 ^$^
Hypertension, yes	4022 (55.1) ^e^	2797 (58.9) ^d^	368 (56.7)	<0.001 ^$^
Dyslipidemia, yes	2473 (33.9)	1669 (35.2)	250 (36.5)	0.197 ^$^
Diabetes mellitus, yes	1574 (21.6) ^e^	1151 (24.3) ^d^	140 (20.4)	0.001 ^$^
Sedentary time, hour/day	8.1 ± 3.6 ^a,b^	8.8 ± 3.8	9.0 ± 4.0	<0.001 ^#^
Fasting glucose, mg/dL	107.5 ± 25.7 ^a^	108.8 ± 27.5	108.3 ± 30.9	0.029 ^#^
Total cholesterol, mg/dL	180.1 ± 39 ^a,b^	182.4 ± 39.9 ^c^	187.5 ± 40.9	<0.001 ^#^
Triglycerides, mg/dL	129.3 ± 73.7	130.8 ± 77.9	133.7 ± 110.2	0.277 ^#^
Total energy intake, kcal	1659.0 ± 658.5 ^a,b^	1557.5 ± 670.3	1556.1 ± 735.2	<0.001 ^#^
Carbohydrate intake, g	284.8 ± 112.2 ^a,b^	266.5 ± 112.2	258.1 ± 124.2	<0.001 ^#^
Protein intake, g	56.1 ± 27.8 ^a,b^	52.4 ± 28.8	53.1 ± 30.5	<0.001 ^#^
Fat intake, g	28.3 ± 21.3 ^a^	26.8 ± 22.3	28.9 ± 23.6	<0.001 ^#^
Cholesterol intake, mg	147.6 ± 158.0	142.0 ± 163.7 ^c^	158.6 ± 175.9	0.017 ^#^
Saturated fatty acid intake, g	8.2 ± 6.7	8.0 ± 7.0 ^c^	8.8 ± 7.6	0.003 ^#^
Dietary fiber intake, g	26.7 ± 14.8 ^a,b^	24.2 ± 14.4 ^c^	22.3 ± 14.1	<0.001 ^#^

Values are presented as mean ± SD or n (%). ^#^, *p*-values for continuous variables were obtained using one-way ANOVA. Superscript letters denote post hoc comparisons: ^a^, high vs. moderate; ^b^, high vs. low; ^c^, moderate vs. low. ^$^, *p*-values for categorical variables were obtained using Pearson’s chi-square test. ^d^, Statistically significant association by adjusted standardized residual > 1.96 (*p* < 0.05). ^e^, Statistically significant association by adjusted standardized residual < −1.96 (*p* < 0.05).

## Data Availability

This study analyzed data released from government agencies: [https://knhanes.kdca.go.kr] (accessed on 15 June 2025).

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
