# Peer review of "Composite Meal-Related Behaviors and Health Indicators: Insight from Large-Scale Nationwide Data on Korean Adults"

_nutrients, 2025, doi:10.3390/nu17182982_

Round 1

Reviewer 1 Report

Comments and Suggestions for Authors

Examining composite dietary behaviors and their associations with health indicators across the lifespan provides valuable insights into patterns of healthy diets. The study is well set up and the conclusions carefully set into perspective.

You can further strengthen the impact of the publication by comparing the approach with other approaches. A short reference is recommended.

Author Response

We sincerely thank the reviewers for their valuable and constructive comments on our manuscript. We have carefully addressed each comment point-by-point, and our detailed responses are provided in the attached file.

Reviewer 2 Report

Comments and Suggestions for Authors

The manuscript entitled “Composite Dietary Behaviors and Their Associations with Health Indicators Across the Adult Lifespan: A Nationally Representative Study and Implications for Healthy Aging” presents interesting issues however some questions arise

  • The article title is quite long and could possibly be shortened.
  • “Participants were classified into three categories…” – it would be helpful to specify on what reference(s) this classification was based. The statement that “Very Good (all three healthy behaviors: regular breakfast, shared mealtimes, and infrequent eating out)” cannot be considered as meeting all criteria for a very good dietary pattern, because it does not include information e.g. on fruit and vegetable intake, fat quality, etc. It primarily reflects behaviors related to meal consumption rather than dietary quality. This distinction should be clearly stated in both the title and the definition of ‘Dietary Behaviors’ or “meal-related behaviors” to avoid misleading readers.
  • Some elements in the abstract could be clarified (e.g., “Older adults,” “Young adults” – specify the exact age ranges).
  • The conclusions in the abstract are very general and do not directly reflect the study results.
  • In the introduction, the authors describe one element of dietary behaviors as “breakfast consumption, shared mealtimes, and frequency of eating out.” It should be clearly emphasized that these are not all components of dietary behaviors. Many aspects of dietary behaviors are not assessed in this study, so it is not accurate to use the term “dietary behaviors” when the study actually examines only meal-related behaviors.
  • “Understanding dietary behaviors across the life course is essential for identifying modifiable factors that influence healthy aging.” – This statement is important; however, given the cross-sectional nature of the study (data collected at a single time point), it is not possible to infer causal effects on healthy aging. The authors note that dietary behaviors may change with age due to shifting demands related to education, work, or social activities. Therefore, cross-sectional data cannot show the impact on healthy aging, because we do not know whether individuals who eat healthily at one point in time have always done so or have changed their behaviors over time. These should be clearly reflected in the study’s aims and in the article title.
  • “This life course approach provides insights into how dietary behaviors evolve with age and identifies potential areas for intervention to support healthy aging.” – Based on the data description, although the study was conducted from 2014 to 2022, data were collected once per participant. Therefore, no conclusions can be drawn about changes over time. If the study was longitudinal (multiple waves), this should be explicitly stated. If it is cross-sectional, the conclusions about changes over the life course must be revised because they are currently unsupported.
  • “2.2. Categorization based on dietary behaviors” – This section needs more detail. The authors should clearly explain the basis (references) for defining the categories (very good, good, and poor) and their components (only three: breakfast consumption, shared meals, and frequency of eating out).
  • They should be presented based on a clear classification (please provide references), into young adults, middle-aged adults, and older adults, and the specific age ranges for each group should be provided. I see this age range only in Figure 2, but it should also be clearly stated in the Methods section.
  • Table 2 – In addition to Body Mass Index (kg/m²), it would be useful to include the number of individuals with normal versus overweight. Considering differences in total energy intake, this could provide an interesting perspective.
  • “Continuous variables were presented as means and standard deviations (SD), and 169 categorical variables were expressed as counts and percentages” - Was the normality of the distribution tested? Information on this should be provided, and the authors should remain consistent in their approach. If the data follow a normal distribution, they should be analyzed accordingly; if not, nonparametric tests should be applied. Please clarify this point.
  • For the multinomial logistic regression analyses, Figure 2 and the entire analysis are difficult to interpret – perhaps presenting the results in a tabular format would be easier. However, this is just a suggestion.
  • The manuscript addresses an important topic on meal-related behaviors and their associations with health indicators across adulthood. However, the study’s cross-sectional design, limited scope of assessed dietary behaviors, and unclear classification criteria limit the strength and clarity of the conclusions. Overall, the paper would benefit from clarifying terminology (specifying age groups; classified three categories of very good, good, poor) add appropriate references, and ensuring that the findings are accurately framed in relation to the study design.

Author Response

(The authors gave the same response as above.)

Reviewer 3 Report

Comments and Suggestions for Authors

Interesting idea of ​​this study, my recommendations are the following:
Abstract – I recommend mentioning the period in years, when the information was collected. I recommend mentioning the parameters targeted in the study.
Lines 63-79 I recommend mentioning the bibliographic sources that support the statements.
I recommend expanding the Introduction section by presenting certain specific aspects regarding the eating behaviors of the Korean population, based on scientific sources.
Methods – I recommend introducing a new subsection called Study design where the typology of the present study and other specific aspects should be mentioned, if applicable. I recommend mentioning whether it is about South or North Korea, or both.
Results – I recommend that when interpreting the results, these should not be duplicated in the value information presented in Figure 1, I recommend revising it.
Calculating the number of participants in the three age categories, from tables 1,2,3, row 1, we found some differences, namely for young adults instead of 15455 there are only 12668, for middle-aged adults 21268 /25416 and for old 13396 instead of 15041, as mentioned in lines 107-109, I recommend clarifications.
I recommend expanding the Discussion section by making new concrete correlations between the results of the present study with results from previous studies targeting time spent sitting.
I recommend expanding the limitations by taking into account other parameters that influence eating behavior, for example the level of physical activity, etc.

Author Response

(The authors gave the same response as above.)

Reviewer 4 Report

Comments and Suggestions for Authors

This study presents a valuable contribution to the field of nutritional epidemiology through its multidimensional assessment of dietary behaviors across the adult lifespan. Its integration of nationally representative data with stratified analyses by age group provides a nuanced understanding of how composite dietary patterns relate to sociodemographic, clinical, and biochemical indicators. The use of KNHANES data and the categorization into Very Good, Good, and Poor dietary behavior groups offers a practical framework for identifying vulnerable populations and informing age-specific interventions. However, several points merit further consideration:

1.The study primarily focuses on associations between dietary behavior patterns and health indicators, yet it does not explore the potential therapeutic effects or adverse consequences of specific dietary components. Could the inclusion of longitudinal data or intervention-based findings help clarify the causal relationships and potential toxicities of these dietary patterns?

2.The reliance on a single 24-hour dietary recall and self-reported behavioral data may limit the generalizability of the findings. Would including additional cases or similar instances from other national cohorts or longitudinal studies enhance the robustness and external validity of the conclusions drawn?

3.The paper suggests that older adults with poor dietary patterns exhibit nutritional deficiencies and social vulnerabilities. Could further exploration and validation of this hypothesis through qualitative assessments, functional status evaluations, or biomarker-based nutritional profiling strengthen the conclusions regarding aging-related risks and intervention targets?

4.While the study identifies living alone and sedentary time as consistent predictors of poor dietary behavior, it does not delve into the potential mediating or moderating roles of psychosocial factors such as depression, social support, or cognitive function. Would incorporating such variables provide a more comprehensive understanding of the behavioral determinants?

5.The discussion highlights the paradox of older adults maintaining regular meal patterns yet experiencing low rates of shared meals. This observation is compelling but underexplored. Could the authors elaborate on potential cultural, economic, or environmental factors that may influence this discrepancy, and suggest targeted strategies to address social isolation in nutritional interventions?

6.The manuscript acknowledges the limitations of its cross-sectional design and self-reported data. However, the implications for policy and practice are presented with considerable confidence. Would a more tempered interpretation, emphasizing the need for longitudinal and interventional studies, better align with the methodological constraints?

Author Response

(The authors gave the same response as above.)

Round 2

Reviewer 2 Report

Comments and Suggestions for Authors

I would like to sincerely thank the authors for the detailed response to my comments and, most importantly, for the valuable revisions made to the article. In this version, the paper is much clearer and easier to follow. The introduction of BMI categories based on the Asian–Pacific guidelines provides a more accurate characterization of this population and allowed the authors to present their findings from this perspective. These categories differ from those proposed by the WHO, but the approach chosen by the authors is appropriate and justified. In summary, the article offers a very interesting contribution to understanding the factors influencing meal-related behaviors in South Korea.

Reviewer 3 Report

Comments and Suggestions for Authors

no comments